



# Assimilation of GNSS Tropospheric Gradients into the Weather Research and Forecasting Model Version 4.4.1

Rohith Thundathil[1,2], Florian Zus[2], Galina Dick[2], and Jens Wickert[1,2]

[1]Institute of Geodesy and Geoinformation Science, Technische Universität Berlin, 10623 Berlin, Germany
[2]GFZ German Research Centre for Geosciences, 14473 Potsdam, Germany

**Correspondence:** Rohith Thundathil (r.thundathil@tu-berlin.de)

**Abstract.** In this study, we have incorporated tropospheric gradient observations from a Global Navigation Satellite Systems (GNSS) ground station network into the Weather Research and Forecasting (WRF) model through a newly developed observation operator. The experiments are aimed to test the functionality of the developed observation operator and to analyze the impact of tropospheric gradients in the sophisticated Data Assimilation (DA) system. The model was configured for a 0.1-degree mesh over Germany with 50 vertical levels up to 50 hPa. Our initial conditions were obtained from the National Centers for Environmental Prediction (NCEP) Global Forecast System (GFS) data at 0.25-degree resolution, and conventional observations were obtained from the European Centre for Medium-Range Weather Forecasts (ECMWF), restricted to surface stations and radiosondes. We selected approximately 100 GNSS stations with high data quality and availability covering Germany. We performed DA every 6 hours for June and July 2021. Three experiments were conducted: 1) The control run assimilating only conventional observations; 2) the impact run assimilating Zenith Total Delays (ZTDs) on top of the Control run; and 3) the Impact-Gradient run assimilating ZTDs and gradients on top of the Control run. The error for the Impact run was reduced by 32% and 10% for ZTDs and gradients, whereas the error for the Impact-Gradient run was reduced by 35% and 18%, respectively. Overall, the newly developed operator for the WRF DA system works as intended. In particular, the combined assimilation of gradients and the ZTDs led to a notable improvement in the humidity field at altitudes above 2.5km. With the source codes developed and freely available to the WRF users, we aim to trigger further GNSS tropospheric gradient assimilation studies to refine the technique and improve its performance.

## 1 Introduction

Water vapor, one of the vital components in the atmosphere, plays a crucial role in weather forecasting and climate research. It is the most abundant greenhouse gas, which accounts for 70% of atmospheric warming and plays a vital role in energy exchange within the atmosphere. However, more knowledge is needed about the humidity field due to limited observations and sub-optimal data assimilation systems. To ensure effective weather forecasting, it is imperative to have precise and uninterrupted observations that serve as the basis for initializing numerical weather prediction models. The spatio-temporal distribution of water vapor information is crucial for accurately modeling the atmosphere. This is especially important for predicting heavy precipitation and severe weather events, which are among the most critical challenges in weather research.





Global Navigation Satellite Systems (GNSS) have profoundly transformed how we determine our position, navigate, and keep track of time. Apart from positioning and timing applications, GNSS is a powerful and versatile tool for geosciences. It can accurately sense atmospheric temperature, water vapor content, ionospheric electron content, Earth surface properties, deformation, and other geophysical parameters (Wickert et al., 2020).

    A crucial aspect of geophysics involves monitoring atmospheric water vapor using GNSS regional ground networks. This

helps to fill gaps in established meteorological observing systems. For instance, Germany currently operates around 270 stations. No other observation network has such a high temporal and spatial resolution. However, there is room for improvement in the impact of the currently provided Zenith Total Delay (ZTD) data products on forecast systems due to limited atmospheric information content.

    GNSS satellites transmit radio signals that ground-based stations receive to estimate ZTDs and tropospheric gradients. This

capability was initially demonstrated by Bevis et al. (1992) for ZTDs and by Bar-Sever et al. (1998) for tropospheric gradients. GNSS meteorology relies on ZTD, which is the core observable strongly correlated with Integrated Water Vapor (IWV) above the station. GNSS stands out among other observation systems for its numerous benefits, including low operating expenses, all-weather availability, and exceptional spatio-temporal resolution. ZTD data is readily accessible from multiple station networks in Europe, such as the European Meteorological Network GNSS Water Vapor Program (E-GVAP), in near real-time. E-GVAP

was established in April 2005 to provide near-real-time GNSS delay data. This data contains information about the amount of water vapor above the GNSS sites. The meteorological data provided by E-GVAP can validate GNSS delay estimation and enhance GNSS positioning in the future. Currently, the E-GVAP network comprises over 3,500 GNSS sites. Studies on assimilation have demonstrated that using ZTD data enhances forecast accuracy, as shown in the findings of Vedel and Huang (2004). Poli et al. (2007) utilized 4DVar to assimilate GNSS-ZTD data into the ARPEGE global model, positively impacting the

assimilation of synoptic-scale circulations and precipitation forecasting during spring and summer. Further research in France by Boniface et al. (2009) and Yan et al. (2009) validated the beneficial influence of incorporating GNSS-ZTD data assimilation into NWP forecasting. Lindskog et al. (2017) performed GNSS-ZTD data assimilation into the HARMONIE-AROME model at a 2.5 km horizontal resolution, indicating that incorporating GNSS ZTD as an additional observation type improves forecast quality and highlights the potential for further improvements in data assimilation through the integration of GNSS ZTD with

other observation types. In a study by Rohm et al. (2019), GNSS data was incorporated into the WRF model at a 4 km horizontal resolution over Poland for two months, revealing a significant enhancement in the model's ability to forecast both water vapor and precipitation. Studies by Giannaros et al. (2020) and Caldas-Alvarez and Khodayar (2020) demonstrate the significant benefits of GNSS-ZTD data assimilation in improving precipitation and water vapor forecast accuracy, focusing on Greece and a broader Mediterranean and Central European region, respectively. Lagasio et al. (2019) discovered that incorporating

diverse Sentinel-1 and GNSS-ZTD observations into the WRF model offers the most significant advantages to forecasts by providing details on the wind field and water vapor content. Mascitelli et al. (2019, 2021) successfully used the RAMS@ISAC model in two experiments to assimilate GNSS-ZTD and IWV data. GNSS-ZTD was assimilated using 3DVar, while IWV was assimilated through nudging. This assimilation led to a notable improvement in short-term water vapor prediction while having a minor effect on precipitation forecasts in both instances.





Several European weather agencies assimilate Zenith Total Delay (ZTD) data. However, its utility is limited because it does not provide information on horizontal or vertical atmospheric gradients (Bennitt and Jupp, 2012; Mahfouf et al., 2015). Tropospheric gradients have the potential to offer valuable additional insights. Until now, most studies have primarily focused on validating tropospheric gradients. Bar-Sever et al. (1998) initially reported preliminary findings confirming genuine atmospheric features in tropospheric gradients. However, their study compared data from only one station to tropospheric gradient

estimates obtained from a collocated Water Vapor Radiometer (WVR) for assessing tropospheric gradients. Subsequently, the exploration of tropospheric gradients gained traction in meteorology. One of the early efforts was by Walpersdorf et al. (2001), who compared tropospheric gradients derived from GPS with those obtained from a Numerical Weather Model (NWM) for a specific set of stations. Iwabuchi et al. (2003) demonstrated a strong correlation between tropospheric gradients and the moisture field. Typically, tropospheric gradients, when plotted as vectors, indicate the direction from dry to moist regions, as

noted by Brenot et al. (2013) in their investigation of deep convection. Li et al. (2015) showed that improving the observation geometry leads to more accurate estimations of tropospheric gradients. Morel et al. (2015) conducted a study on tropospheric gradients using various software packages, analyzing data from twelve stations on Corsica, a Mediterranean island. However, it's noteworthy that these studies often involve a limited number of stations. Douša et al. (2016) compared multiple stations and two GNSS analysis techniques with tropospheric gradients from NWMs. Their visual inspection of tropospheric gradient

maps provided compelling evidence that GNSS tropospheric gradients accurately capture actual tropospheric features. While they made positive findings, certain aspects required further exploration, such as understanding the role of GNSS data processing options and parameters. During the 2nd EUREF reprocessing, Dousa et al. (2017) identified distinct artificial signals in tropospheric gradients, attributed to absorption of asymmetric effects resulting from instrumentation issues. They also noted seasonal variations in deviations between GNSS and NWM ZTDs and tropospheric gradients, with larger deviations observed

in summer due to NWMs struggling to predict high water vapor variability. The data suggests that while the standard deviation for ZTD remained consistent over time, the standard deviation for tropospheric gradients decreased, indicating improved quality over the years. Since their introduction in the early '90s, GNSS ZTDs have consistently maintained high quality in meteorology. Kačmařík et al. (2019) studied the sensitivity of tropospheric gradients to various processing options and found that post-processing mode solutions reliably estimated tropospheric gradients correlating well with actual weather conditions.

The accuracy of real-time tropospheric gradient estimates primarily depends on the availability of high-quality satellite orbits and clocks. The purpose of this study is to advance a step further by making use of tropospheric gradients. One possibility is to assimilate tropospheric gradients into an NWM, requiring the development and implementation of observation operators into data assimilation systems. This research incorporated a new observation operator in the WRFDA system, which allows one to assimilate GNSS tropospheric gradients.

The developed operator is an upgrade or an add-on to the current ZTD operator already implemented in the WRFDA. Hence, the modules connected with the ZTD operator codes were modified with additional code snippets to create the ZTD+gradient version of the operator. Users can quickly assimilate gradients with this operator by adding a few lines to the WRFDA namelist.input. The process can be controlled with switches in the namelist, making it a hassle-free task for anyone interested in assimilating gradient observations. The codes will be included in the manuscript's supplementary material and later





uploaded to the National Center for Atmospheric Research (NCAR) for incorporation in future official versions. This research project titled "Exploitation of GNSS Tropospheric Gradients for Severe Weather Monitoring and Prediction (EGMAP)" is funded by the German Research Foundation (DFG). EGMAP aims to optimize the utilization of GNSS tropospheric gradients to predict severe weather for operational purposes.

The manuscript will commence by a comprehensive overview of the GNSS ZTD and gradient operators. It will then provide
a detailed description of the model domain and the DA system, including the assimilation datasets. The results section will present the single observation tests (SOT) of the gradients and compare it to the ZTD observation. The manuscript will then conduct a qualitative analysis of the assimilation impact, followed by a quantitative analysis demonstrating improvements in the humidity fields. In order to validate these findings, our study will utilize ERA5 datasets and radiosondes. Finally, the paper will conclude with a summary.

## 2    GNSS ZTDs and Tropospheric Gradients

The study makes use of two types of observations: ZTDs and tropospheric gradients. ZTD assimilation is a well-researched field currently operational in weather models and is used by various forecasting agencies. However, tropospheric gradients have not been assimilated into weather models until today. This research focuses on assimilating the so-called East and North gradient components into the WRF model. This is achieved through our newly developed forward operator, which has been
implemented in the latest version of WRF. Initially, we demonstrate how ZTDs and tropospheric gradients are obtained from ground-based stations.

The signal travel time delay caused by the neutral atmosphere is parameterized in the GNSS analysis. The tropospheric delay ($T$) at the receiving station is expressed as a function of the elevation angle ($e$) and azimuth angle ($a$)

$$T(e,a) = m_h(e).Z_h + m_w(e).Z_w + m_g(e)\left[\cos(a).N + \sin(a).E\right] \tag{1}$$

where $Z_h$ represents the zenith hydrostatic delay (ZHD), $Z_w$ is the zenith wet delay (ZWD), and $N$ and $E$ denote the north and east gradient components. The hydrostatic, wet, and gradient Mapping Functions (MFs) are denoted as $m_h$, $m_w$, and $m_g$, respectively. The ZTD, $Z$, is given by

$$Z = Z_h + Z_w. \tag{2}$$

In GNSS analysis, the ZTD, along with the north gradient component $N$ and the east gradient component $E$, are jointly
estimated with geodetic parameters through a least squares adjustment as described in Gendt et al. (2004). These three quantities are treated as observations, and their assimilation necessitates the development of forward operators. Essentially, we need to establish a method for computing the ZTD, north gradient, and east gradient components within the weather model. This process is detailed in the following section.





## 2.1 The ZTD and tropospheric gradient operator

The ZTD is calculated through

$$ZTD = 10^{-6} \int \Psi.dz \qquad (3)$$

The refractivity $\Psi$ is a function of pressure, temperature, and humidity (Thayer, 1974), with $z$ denoting the height above the station. The forward operator for the ZTD, along with the tangent-linear and adjoint operators, is already integrated into the WRFDA system. For more details, please refer to the three respective routines in the WRFDA code.

1. In 'da_get_innov_vector_gpsztd.inc,' the innovation, i.e., the difference between the observed and forward-modeled ZTD, is calculated.

2. The routine 'da_transform_xtoy_gpsztd.inc' represents the corresponding tangent-linear code, while,

3. the routine 'da_transform_xtoy_gpsztd_adj.inc' represents the corresponding adjoint code.

Regarding tropospheric gradients, we have identified two possible approaches for the forward operator, which we refer to 135 as the rigorous and fast methods. The rigorous approach involves the following steps: We compute numerous tropospheric delays, considering various elevation and azimuth angles for the given station location, and then perform a least squares fit to obtain the north and east gradient components (as described in Zus et al. (2018)). The rigorous approach aims to replicate how tropospheric gradients are estimated in GNSS analysis. However, it has a drawback—it can be challenging to implement in data assimilation (DA) systems due to the need for a ray-tracing algorithm (Zus et al., 2015).

Therefore, we prefer the fast approach, which works as follows: For the given station location, we utilize a closed-form expression that depends on the north-south and east-west horizontal gradients of refractivity (as outlined in Davis et al. (1993)). This enables the calculation of the north and east gradient components through

$$N = 10^{-6} \int z.\Psi_y.dz \qquad (4)$$

$$E = 10^{-6} \int z.\Psi_x.dz \qquad (5)$$

Here, $x$, $y$, and $z$ represent Cartesian coordinates, and the subscripts denote partial derivatives. Therefore, similar to ZTD, the tropospheric gradients are computed through numerical integration. In essence, we define a sequence of integration points, calculate the integrant at these points, and then sum them up along with their respective interpolation weights (as described in Zus et al. (2012)). The critical aspect lies in the computation of the horizontal refractivity gradients, $\Psi_y$ and $\Psi_x$, at the 150 location of these integration points. To start, we express these gradients in terms of the station's longitude ($\lambda$) and latitude ($\phi$) as follows:

$$\Psi_y = \frac{\Psi_\phi}{\rho} \qquad (6)$$





$$\Psi_x = \frac{\Psi_\lambda}{\rho . \cos\phi} \tag{7}$$

where $\rho$ represents the radial distance or the distance to the center of the osculating sphere. Our proposed method for calculating the horizontal refractivity gradients $\Psi_\phi$ and $\Psi_\lambda$ involves a least squares adjustment. Essentially, we establish a relationship between the refractivity $\Psi$ and the horizontal refractivity gradients $\Psi_\phi$ and $\Psi_\lambda$ at a specific (geometric) height $h$, and the vertically adjusted refractivity at neighboring grid points of the weather model using Taylor's series expansion.

$$\Psi_1 = \Psi + (\lambda_1 - \lambda_0).\Psi_\lambda + (\phi_1 - \phi_0).\Psi_\phi$$

$$\Psi_2 = \Psi + (\lambda_2 - \lambda_0).\Psi_\lambda + (\phi_2 - \phi_0).\Psi_\phi$$

$$\Psi_3 = \Psi + (\lambda_3 - \lambda_0).\Psi_\lambda + (\phi_3 - \phi_0).\Psi_\phi$$

$$...$$

$$\Psi_m = \Psi + (\lambda_m - \lambda_0).\Psi_\lambda + (\phi_m - \phi_0).\Psi_\phi \tag{8}$$

Through vertical interpolation, the vertically adjusted refractivity $\Psi_i$ can be determined by

$$\Psi_i = \Psi_i^k + \frac{\Psi_i^{k+1} - \Psi_i^k}{h_i^{k+1} - h_i^k}(h - h_i^k) \tag{9}$$

Above the weather model top, the vertically adjusted refractivity $\Psi_i$ is determined using the hydrostatic equation

$$\Psi_i = \Psi_i^n . \exp[-G\frac{h - h_i^n}{T_i^n}] \tag{10}$$

where $\Psi_i^n$, $T_i^n$, and $h_i^n$ represent the respective refractivity, temperature, and height at the top of the weather model, and G is the hydrostatic constant. In this equation, we approximate the geopotential height by the geometric height. To obtain the horizontal refractivity gradients $\Psi_\phi$ and $\Psi_\lambda$, we invert the linear system of 8 using the least squares method. Consequently, the horizontal refractivity gradient at a specific height can be expressed as a linear combination of the vertically adjusted refractivity $\Psi_i$.

$$\Psi_\phi = \sum_{i=1}^{m} w_i . \Psi_i \tag{11}$$


$$\Psi_\lambda = \sum_{i=1}^{m} v_i . \Psi_i \tag{12}$$

where $w$ and $v$ represent the interpolation weights. The least-squares adjustment incorporates grid points within a 35 km radius to compute the horizontal refractivity gradients. For example, assuming the weather model has a 10 km horizontal resolution, this computation involves 8 x 8 grid points. The selection of grid points included in the least squares fit is not arbitrary; it is

justified by comparing tropospheric gradients derived from the rigorous and fast approaches (as discussed in Zus et al. (2023)).

The tangent-linear operator is derived by applying the chain rule of differential calculus in forward mode. Similarly the adjoint operator is derived by applying the same in the reverse mode (Giering and Kaminski, 1998). However, two simplifications





are introduced. First, the geometric height, derived from the geopotential height (the 'natural' height of the weather model), is not treated as a control variable. In other words, the partial derivatives in the construction of the tangent-linear (adjoint) code

are ignored. This simplification is also applied to other operators, such as the operator for radio occultation refractivity profiles. Second, the temperature we use to compute the refractivity above the weather model's top is not treated as a control variable. This is not problematic because tropospheric gradients are not very sensitive to refractivity at high altitudes (Zus et al., 2018).

The newly developed operator is not implemented as a stand-alone entity. Instead, we integrated the tropospheric gradient operator as an add-on to the existing ZTD operator. For each respective station, the code for tropospheric gradients is automat-

ically executed alongside the ZTD code. To control the assimilation of tropospheric gradients, users can use a single variable in the namelist called 'use_gpsgraobs' to switch it on or off. Therefore, for in-depth information about the forward operator, readers are directed to the routine 'da_get_innov_vector_gpsztd.inc.' Details on the tangent-linear and adjoint operators can be found in the routines 'da_transform_xtoy_gpsztd.inc' and 'da_transform_xtoy_gpsztd_adj.inc,' respectively.

I would like to note that the MPI implementation of the newly developed operator presents a unique challenge. While

the existing ZTD operator is considered 'local' because it utilizes variables from nearby (four surrounding) grid points, the tropospheric gradient operator is considered 'non-local' due to the inclusion of numerous nearby grid points when calculating horizontal refractivity gradient components. This necessitates adjustments to the size of the 'halo' region. Instead, we opted to implement an approach similar to the one used for the non-local excess phase path operator (Chen et al., 2009). This approach ensures that the global refractivity (and temperature) field is available to individual processors. For more in-depth information,

readers are encouraged to consult the WRFDA system documentation.

## 3 Model setup

### 3.1 WRF model and configuration

We employed the non-hydrostatic WRF model version 4.4.1 for our impact study, implementing the gradient observation operator. WRF has a strong track record in both the research community and operational forecasting agencies worldwide,

making it an ideal platform to test our gradient operator. We specifically used the Advanced Research WRF (ARW) core (Skamarock et al., 2008).

The model domain (Figure 1) was configured with a 0.1 ° (≈ 11 km) horizontal resolution, featuring a grid of 200 x 200 points. Vertically, the model includes 50 levels, extending to a model top of 50 hPa. We set the model time step for the WRF forecast simulation at 30 seconds. Our model forecast simulations were driven by the National Centers for Environmental

Prediction (NCEP) Global Forecast System (GFS) operational analysis data, with a spatial resolution of 0.25 degrees (approximately 27 km).

For our model physics, we used the following parameters, as listed in Table 1. The radiation parameterization included the RRTMG scheme (Iacono et al., 2008), which is known for accurately and efficiently calculating long-wave and short-wave fluxes and heating rates, particularly suited for General Circulation Model (GCM) applications. The cloud microphysics

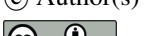



scheme was the Thompson double-moment scheme (Thompson et al., 2008), capable of predicting mixing ratios for cloud water, rain, ice, snow, and graupel.

Our planetary boundary layer (PBL) scheme of choice for this simulation was the YSU scheme (Hong et al., 2010; Hong and Lim, 2006). The YSU scheme, a nonlocal scheme with first-order closure, incorporates counter-gradient and explicit entrainment terms in the turbulence flux equation.

We utilized the unified Noah land surface model (Chen and Dudhia, 2001) for this study. This model comprises four layers and predicts soil temperature and moisture, canopy moisture, and snow cover. It considers various factors, including root zone dynamics, evapotranspiration, soil drainage, runoff variables, vegetation categories, and soil texture. This comprehensive approach provides valuable information on sensible and latent heat fluxes related to the boundary layer and incorporates an enhanced urban treatment.

To accurately simulate the model at a non-convective scale resolution, it is essential to include convection parameterization. This helps represent the statistical impact of sub-grid-scale convective clouds. For this purpose, we employed the Grell-Freitas ensemble scheme (Grell and Freitas, 2014), which combines a probability density function with data assimilation techniques.

## 3.2 Data assimilation system

The DA systems in WRF are typically classified into three categories: deterministic, probabilistic, and hybrid systems, which

combine elements from both. In this study, we utilized the deterministic three-dimensional variational (3DVAR) DA system.

In the 3DVAR DA system, the objective is to iteratively minimize the cost function $J(x)$, where the independent or control variable is the analysis state vector $x$. The cost function for the 3DVAR system is represented as follows:

$$J(x) = \frac{1}{2}(x - x_b)^T \mathbf{B}^{-1}(x - x_b) + \frac{1}{2}(y - \mathbf{H}(x))^T \mathbf{R}^{-1}(y - \mathbf{H}(x)) \tag{13}$$

The cost function $J(x)$ consists of two main terms: a background term and an observation term. The variables $x$, $x_b$, and $y$

are column vectors representing the analysis state, the background (or first guess), and the observation state, respectively. The forward operator, denoted as $\mathbf{H}$, is responsible for mapping the analysis state vector space to the observation vector space. In this study, we have implemented the forward operator $\mathbf{H}$ specifically for GNSS tropospheric gradients.

Alongside the column vectors, there are two square matrices that play a crucial role in minimizing the cost function: the background error covariance matrix $\mathbf{B}$ and the observation error covariance matrix $\mathbf{R}$. $\mathbf{R}$ is a diagonal matrix because we assume

that observation errors from various sources are uncorrelated. On the other hand, $\mathbf{B}$ is a square, positive semi-definite, and symmetric matrix with positive eigenvalues. It includes variances of background forecast errors on the diagonal and covariances between them on the symmetric upper and lower triangular elements. After assimilating an observation, the variances and covariances in $\mathbf{B}$ significantly influence the analysis response. Therefore, accurately determining $\mathbf{B}$ is essential in a variational DA system.

In this research, we computed the $\mathbf{B}$ matrix using the National Meteorological Center (NMC) method (Parrish and Derber, 1992). The NMC method is widely used for generating B by estimating climatological background error covariances. We selected the NMC method due to its ability to yield physically sound results within regional model domains and its lower





computational cost compared to ensemble methods. The NMC method involves calculating forecast difference statistics to obtain the forecast error covariance tailored to a specific domain. However, it does have limitations, such as overestimating

covariances in large-scale simulations and regions with poor observation (Berre, 2000; Fischer, 2013; Berre et al., 2006).

For our regional simulations, forecast statistics were derived from analyzing forecast differences over a month, utilizing both 24-hour and 12-hour predictions. These statistics were obtained from data in June 2021. We chose the CV5 option as it allows independent control of moisture levels without interference from other variables.

### 3.3 Experimental setup

The assimilation experiment was conducted for two months, in June and July 2021, using a rapid-update cycle (RUC) approach, as illustrated in Figure 2. The RUC was configured for 6-hourly DA cycles. The datasets employed for assimilation included: 1) Conventional data, comprising surface stations (SYNOP) and radiosondes, 2) Zenith Total Delays (ZTDs), and, 3) Tropospheric gradients.

We performed three primary experiments for this study:1) A Control run, incorporating only conventional observations

(SYNOP and radiosondes), 2) an Impact run (ZTD), assimilating ZTD observations in addition to the Control run, and, 3) an Impact Gradient run (ZTDGRA), where both ZTD and gradient observations were assimilated on top of the Control run, using the newly developed gradient operator. The assimilation period spanned from 1st June 00 UTC to 31st July 18 UTC, 2021, with six-hour intervals, totaling 244 DA cycles. The initial DA cycle commenced after a 12-hour spin-up run aimed at stabilizing the model's initial and boundary conditions, ensuring reliable forecasts for subsequent DA.

### 3.4 Datasets for assimilation

#### 3.4.1 Conventional datasets

To enhance the capabilities of the DA system, we established a comprehensive network of surface reports, SYNOP, across Europe. Radiosonde (RS) measurements, obtained through TEMP, provide a detailed view of the atmospheric thermodynamic structure at launch points. These valuable observations are conveniently accessible via the World Meteorological Organization's

(WMO) Global Telecommunication System (GTS) data archive, which is housed at ECMWF.

To maintain simplicity within the DA system, we limited the use of conventional datasets to surface observations and radiosondes. It's worth noting that the primary objectives of this work are as follows:1) To test the functionality of the newly developed operator (code), and, 2) to analyze the difference between the experiment where we assimilate ZTDs only and the experiment where we assimilate both ZTDs and tropospheric gradients. In essence, our focus lies on assessing the relative

impact rather than the absolute impact of GNSS data in variational DA.

#### 3.4.2 GNSS ZTDs and tropospheric gradient observations

For detailed information regarding the GNSS analysis conducted at GFZ with its in-house software package EPOS (Earth Parameter and Orbit Determination System), readers are encouraged to refer to works such as Gendt et al. (2004).





Initially, we had access to approximately 380 globally distributed stations. However, we refined our dataset by removing
stations located outside the considered domain and those with data availability below 75%. Subsequently, we evenly distributed
the remaining stations across Germany. This approach resulted in an assimilation dataset comprising slightly more than 100
stations, covering Germany and parts of its borders.

To address potential biases in the GNSS dataset, we relied on analyses from our control experiment for bias correction.
We utilized the two-month simulation data from the control experiment to perform station-specific bias correction for GNSS
ZTDs and gradient observations, which were then applied in the ZTD and ZTDGRA experiments. We assigned an observation
error of 8 mm for the ZTD and 0.65 mm for the gradient for all observations. The observation error for the ZTD is motivated
by previous assimilation studies. The choice of observation error for the tropospheric gradient components is informed by an
analysis of the Observation minus Background (OB) statistics from the control experiment (see results below). It's important
to note that the OB statistics represent a composite of observation and model errors, suggesting that our current choice for the
observation error may be somewhat 'pessimistic.' In future work, we plan to conduct sensitivity studies to obtain more accurate
estimates for observation errors.

## 4 Results

### 4.1 Single observation tests

To assess the impact of assimilating a single observation from a station location, we conducted the Single Observation Test
(SOT) in WRF. Through the SOT, our aim was to gain insights into the model's behavior in response to GNSS gradient
observations. The DA system employed here is the 3DVAR, where the extent of the assimilation impact largely depends on the
B matrix. We sought answers to the following questions:

1. How does the impact region differ when compared to assimilating ZTD observations?

2. What distinguishes the SOTs resulting from the assimilation of gradients from those involving ZTD observations?

3. How sensitive is the model to the assimilation of a single gradient observation?

To address these questions, we selected a point at the center of the model domain and conducted SOTs with varying pseudo-
ZTD and gradient observations and their associated errors.

For questions 1 and 2, we conducted separate SOTs for gradients and ZTDs. In the gradient SOT, we selected an increment
of -1.0 mm for the North gradient observation value and 0 mm for the East gradient observation value, with an observation
error of 1.0 mm. In the ZTD SOT, we used an increment of 1.0 cm for the ZTD value and an observation error of 1.0 cm. We
opted for unit values for errors to facilitate the understanding of the increment's impact without a scaling factor.

Figure 3 provides a comparison of the spatial impact resulting from SOTs using gradient and ZTD observations. This
comparison helps us visualize how the impact spreads. The gradient impact plot in Fig. 3a justifies the name 'gradient' as
it reveals increased moisture in the south (positive lobe) and decreased moisture in the north (negative lobe), indicating a





redistribution of moisture in the analysis. Visualizing the gradient increment as a vector pointing south aligns with the input values of -1.0 mm for North gradient and 0 mm for East gradient. Tropospheric gradients can be considered as moisture vectors, similar to wind vectors, encompassing both north-south and east-west components. Typically, tropospheric gradients point from dry to moist areas.

Comparing the spatial impact of SOTs, we observe that the maximum impact response for a -1 mm North gradient increment with a 1 mm error is 0.062 g kg$^{-1}$. In contrast, the maximum response for the SOT due to a 1 cm magnitude ZTD increment with a 1 cm error is 0.2 g kg$^{-1}$. The impact radius reduces by 50% from 0.062 g kg$^{-1}$ to 0.032 g kg$^{-1}$ within a radius of 67 km for the gradient and around 80 km for the ZTD.

Figure 4 displays the Water Vapor Mixing Ratio (WVMR) and temperature profiles of the SOT analysis with respect to model levels. The gradient SOT profile exhibits both positive and negative lobes, explaining the positive and negative responses in the WVMR profile. The temperature response in both gradient and ZTD SOTs is negligible. It is evident that the impact to a large extent depends on the **B** matrix, particularly with the CV5 option, which assumes moisture independence from other variables.

The profile plots highlight that gradient observations have a more significant impact on the lower troposphere than on the surface level, where the impact is minimal. In contrast, ZTD impacts are more pronounced in the lower troposphere but also significant at the surface level. A cross-section in Figure 5 further emphasizes this difference, showing that ZTD impact extends from the lower troposphere to the surface with only a slight decrease in the WVMR value compared to the gradient impact. The gradient impact is most prominent between 1 km and 5.5 km, with minimal influence at the surface level.

To gain a more realistic understanding of the impact of tropospheric gradients, we conducted an SOT using an actual gradient observation from a GNSS station. We selected a station near the center of the model domain with coordinates 52 ° 38 ′ 35 ″ N and 9 ° 12 ′ 22 ″ E. The observation values were 0.497 mm East gradient, 0.099 mm North gradient, with an observation error of 0.65 mm assigned. Unlike previous SOTs that focused on one gradient direction, this scenario involved increments in both the East and North gradient components, resembling real scenarios. In this case, the gradient dipole's direction is the vector sum of the two gradient components. Figure 6 displays the spatial and profile plots of this SOT.

We modified the WRFDA code to allow users to conduct SOTs by adding specific options in the WRF 'namelist.input' file.

## 4.2 Qualitative analysis of the gradient assimilation impact

This section aims to assess the noticeable impact of gradient observations on assimilation, specifically analyzing the spatio-temporal features emerging in the model as a result of assimilating gradients. To illustrate the influence of gradients, we compared the analyses obtained from the ZTDGRA and ZTD experiments through difference plots. The analysis is averaged over vertical levels between 2 km and 4 km to capture the comprehensive impact from the lower troposphere, where gradient observations have the most significant influence.

Assimilation was conducted every 6 hours in a Rapid-Update Cycle (RUC) DA environment. In a RUC DA environment, the initial assimilation impact is of utmost importance as it reflects the analysis increment resulting from the assimilation of fresh observations, localized to their respective locations. As the cycles progress, the features tend to expand on a larger scale, making visual analysis more challenging. Therefore, our focus is primarily on the first and second DA cycles.





The initial DA cycle occurred on June 1, 2021, at 00 UTC. Following this, a 6-hour free forecast was initiated based on the
340 analysis until the next assimilation at 06 UTC. Figure 7 illustrates the initial DA cycle, with Figure 7a showing the analysis
difference observed at 00 UTC, while Figures 7b-f display the forecast differences from 01 UTC to 05 UTC. By comparing
Figure 7a with the station network in Figure 1, it's evident that data from GNSS receivers in the region have had a substantial
impact. Approximately 100 stations located exclusively in Germany have contributed to this impact. The absolute magnitude
of the impact ranges from -0.15 g kg$^{-1}$ to 0.14 g kg$^{-1}$.

345 While the impact of assimilating gradients may seem relatively small in magnitude, it significantly affects the distribution
of the moisture field in the initial model state (Figure 7a). While these figures provide valuable insight, a quantitative analysis
is necessary to determine if the assimilations have successfully corrected the model's moisture fields. Subsequent sections will
delve into this topic, providing detailed comparisons for clarity.

Comparing Figures 7b-f reveals that the impact of the gradients persists in the model for a significant period (6 hours,
350 until the next assimilation). This persistent impact serves as compelling evidence of the reliability of gradient data. Figure 8
reinforces this statement by displaying the analysis of the second DA cycle and the subsequent 6-hour forecast difference. The
structures developed during the forecast difference at 05 UTC are comparable to those observed in the analysis difference at
06 UTC during the second assimilation, as demonstrated by the comparison of Figures 7f and 8a. The structures that emerged
during the second assimilation are not significantly different from the forecast made before the assimilation, indicating that
the gradient observations were accurate, and the assimilation did not degrade the model state neither tampering the flow of the
mositure fields.

## 4.3 Quantitative analysis of the gradient assimilation impact

To comprehensively assess the impact of assimilating GNSS observations, we conducted a quantitative analysis and compared
the results with the original station data from the GNSS network. Our objective was to gauge the extent of improvement
achieved through this assimilation process. Our observation network initially comprised just over 100 GNSS stations. However,
for validation purposes, we deliberately excluded 18 stations, categorizing them as "blacklisted" stations.

These "blacklisted" stations were strategically selected to ensure a balanced spatial distribution, aligning them with the loca-
tions of the German Weather Service (DWD) RADAR stations. They were set aside solely for independent validation purposes.
The remaining GNSS station data, known as the "whitelisted" stations, were assimilated in both the ZTD and ZTDGRA ex-
periments. Consequently, we could evaluate the progress by comparing the assimilated data with the independent observations
from these "whitelisted" stations.

This approach allowed us to rigorously assess the impact of our assimilation efforts and validate the improvements achieved.
We will go through the Root Mean Square Error (RMSE) impact with respect to the whitelisted and blacklisted stations and
also look in to the average impact (in terms of RMSE) as a function of the forecast length in between the assimilation intervals.



### 4.3.1 RMSE with respect to whitelisted stations

To assess the impact of assimilating ZTD and gradient observations on the analyses in observation space, we computed the RMSE for three experiments: 1) Control, 2) ZTD, and 3) ZTDGRA, using data from the whitelisted GNSS stations. We compared the ZTD and gradient values derived from the analyses at the station coordinates with the observations collected by the GNSS stations. This comparison was performed over the entire two-month period, using hourly data, and considering all 375 the whitelisted stations. Figures 9, 10, and 11 present the station-specific RMSE values and the average RMSE for the Control, ZTD, and ZTDGRA experiments, respectively.

From the figures, it is evident that when assessing the improvement in the ZTD variable, the mean RMSE values for the ZTD parameter are the lowest for the ZTDGRA experiment compared to the other runs, providing clear evidence of the successful gradient assimilation impact. The mean RMSE of the ZTD variable for the Control run was 14.4 mm, which reduced to 9.7 380 mm in the ZTD run and further decreased to 9.3 mm in the ZTDGRA run.

Regarding the impact on gradient components, both the North and East components exhibit similar enhancements. Notably, the ZTDGRA assimilation leads to the most significant improvement when compared to the other runs. The North (East) gradient RMSE decreased from 0.68 mm (0.69 mm) in the Control run to 0.61 mm (0.62 mm) in the ZTD run and further decreased to 0.56 mm (0.56 mm) in the ZTDGRA run. The assimilation of gradients has significantly reduced the RMSE 385 values, indicating a substantial enhancement of the moisture field in the model state.

The results shown in Figures 9, 10, and 11 can be interpreted as follows: when assimilating only ZTDs, not only are the ZTDs adjusted, but the tropospheric gradients are also affected (Figure 10). This suggests that tropospheric gradients contain valuable information. Such adjustment would not be possible if tropospheric gradients contained no useful data. However, as long as we do not assimilate tropospheric gradients, we do not make use of their information. This is addressed in the experiment where 390 both the ZTDs and the tropospheric gradients are assimilated (Figure 11). It can be observed that the tropospheric gradients are further adjusted, demonstrating the functionality of our implementation, and that the ZTDs are also further adjusted, providing evidence that we extracted valuable information from the tropospheric gradients.

### 4.3.2 RMSE with respect to blacklisted stations

To further evaluate the consistency of our findings from the whitelisted stations, we conducted a similar analysis using the 395 18 blacklisted stations. We measured the RMSE values as described previously, but specifically for these 18 stations. This comparison was performed using hourly data over the entire two-month period to provide a robust statistical assessment. Figures 12, 13, and 14 present the RMSE comparisons for the Control, ZTD, and ZTDGRA experiments, respectively, using data from the blacklisted stations.

From these figures, it is evident that the RMSE values for the ZTD variable have decreased. The mean RMSE for the ZTD 400 variable decreased from 14.2 mm in the Control run to 10.2 mm in the ZTD run and further reduced to 9.7 mm in the ZTDGRA run.





Similar improvements in the North and East gradient components were observed, mirroring the results obtained with the whitelisted stations. The RMSE values for the North (East) gradient component dropped from 0.68 mm (0.68 mm) in the Control run to 0.62 mm (0.61 mm) in the ZTD run and further decreased to 0.58 mm (0.57 mm) in the ZTDGRA run.

These results clearly demonstrate that the assimilation of gradients has enhanced the model's analyses. The two-month statistics using independent GNSS station data validate the improvements achieved through gradient assimilation.

### 4.3.3   Average impact as a function of forecast length

As our assimilation process occurs every six hours, we inherently have a 6-hour forecast lead time. To calculate the mean RMSE for both ZTD and gradients, considering both whitelisted and blacklisted stations, we examined the mean RMSE values
across forecast lead times ranging from 0 to 5 hours, similar to our previous analysis. Figure 15 illustrates the RMSE variation with respect to forecast length.

   In both whitelisted and blacklisted stations, the ZTDGRA experiment consistently outperforms the other models, displaying the lowest RMSE values. As expected, the RMSE tends to increase as the lead time extends. However, what's noteworthy is the sustained improvement in RMSE across different forecast lead times in both whitelisted and blacklisted stations. This
consistency of improvement provides further evidence of the positive impact of gradients on the model state's moisture field. Additionally, the ZTDGRA experiment maintains a stable RMSE improvement throughout the forecast lead time, indicated by the small but consistent offset.

### 4.4   Comparison with ERA5

In addition to GNSS observation data, we conducted validation using the ERA5 dataset, which is considered one of the most
comprehensive atmospheric reanalyses of the global climate to date, produced by the Copernicus Climate Change Service (C3S) at ECMWF. ERA5 provides hourly estimates of climate variables globally on a 30km grid, offering detailed atmospheric information with 137 levels up to 80km in altitude. Given its high quality, ERA5 serves as a valuable reference dataset. To perform this validation, we selected five locations within the model domain, as shown in Figure 16. With a total of 244 assimilation cycles, each generating five profiles, we had a substantial dataset of 1220 profiles for comparison with ERA5.
This large number of profiles allowed us to obtain robust statistical results.

   We compared the water vapor (WV) profiles for three different lead times after the assimilation cycles: 0, 3, and 5 hours. Figure 17 presents the WV one-sigma deviation between the assimilation experiments and ERA5 as a function of altitude for these three lead times. The findings from Figure 17 indicate that assimilating ZTDs has a positive impact across the entire altitude range. This confirms that the assimilation of ZTDs results in an improvement in the WV field.

Furthermore, the assimilation of tropospheric gradients, in addition to ZTDs, leads to further improvements in the WV field. However, it's important to note that this improvement is primarily restricted to altitudes above 2.5km. Below an altitude of 2.5km, particularly near the Earth's surface, the assimilation of gradients, in addition to ZTDs, does not have a significant impact. This lack of impact close to the Earth's surface is not surprising, as gradients are not sensitive to surface variables (as





indicated by equations 4 and 5). Any impact near the surface can be attributed to the background error covariance matrix and
its correlations between different altitudes, as observed in the Single Observation Test (SOT) experiment.

The positive impact of gradient assimilation remains consistent across different lead times. In essence, the reduction in the
standard deviation above 2.5km observed in the ZTDGRA experiment remains consistent even at the fifth-hour lead time.

### 4.5 Forecast comparison with radiosonde profiles

Forecast validation was carried out for the analysis forecasts generated in all three experiments. The analyses from each of
these experiments were subjected to a free forecast of 12 hours, starting from the DA cycle at 00 UTC on June 1, 2021, and
continuing until 00 UTC on July 30, 2021. We compared the profiles from the 12th hour of these forecasts with radiosonde
profiles, which were also used for assimilation in the model. Since the forecasts began from analyses that were 12 hours
prior, the radiosonde observations provided a perfect basis for comparison. At each time epoch, there were approximately 38
radiosonde observations available.

Figure 18 presents a 60-day time series illustrating the RMSE comparison between the analysis profiles and the radiosondes.
While the improvement observed in the ZTDGRA experiment is notable, it remains relatively modest. It's important to note that
this improvement may not persist beyond the 12th hour of forecasting. Additionally, any non-uniformity among the radiosonde
profiles could potentially impact the final assessment of these improvements.

### 5 Summary

In this study, we successfully implemented a newly developed gradient observation operator within the WRF DA system,
allowing us to assimilate GNSS tropospheric gradients. We utilized WRF version 4.4.1 along with the 3DVAR DA system,
configuring the model with a resolution of 0.1 ° ($\approx$ 11 km) and 50 vertical levels. Our dataset included GNSS ZTDs and
tropospheric gradient observations obtained from over 100 GNSS ground-based stations covering Germany. Assimilations
were performed at six-hour intervals over a two-month period in June and July 2021.

To assess the impact of tropospheric gradients, we conducted three distinct model simulations. The first was a Control
run, which incorporated only conventional observations. The second, the ZTD run, involved assimilating ZTDs in addition to
conventional observations on top of the Control run. Finally, in the ZTDGRA run, we assimilated both ZTDs and gradients
alongside conventional observations on top of the Control run.

Our results clearly demonstrated the positive impact of tropospheric gradient observations when assimilated using the gradi-
ent operator. The ZTDGRA experiment exhibited the smallest RMSE (measured in observation space) compared to the other
experiments, confirming that gradient observations contained valuable information and significantly improved the initial state
of the model. The RMSE for the ZTD run was reduced by 32% for ZTDs and 10% for gradients, while the ZTDGRA run saw
a reduction of 35% for ZTDs and 18% for gradients.

Additionally, our forecasts generated from the analyses exhibited improved accuracy when compared to ERA5 and ra-
diosonde data, suggesting that the gradient information persisted in the model for at least 6 hours. It's important to note that the



most significant impact of gradient observations was observed in the lower troposphere, with negligible effects on the surface level.

While our study focused on the relative impact of gradient assimilation, we acknowledge that further improvements can be achieved through ensemble-based DA systems. Ensemble DA systems utilize flow-dependent background error covariance matrices, considering the dynamic nature of the atmosphere, and are expected to provide a more accurate representation of humidity variables in real-time scenarios. As a next step, we plan to explore GNSS data assimilation using ensemble DA techniques.

With our source codes openly available, we encourage fellow researchers to conduct their own GNSS data assimilation experiments assimilating gradient data and further build upon our findings to advance the field of atmospheric modeling and forecasting.

*Code and data availability.* The complete WRFDA code version 4.4.1 with the gradient operator codes, and the simulation experiment data, including the namelist files and data used for all the assimilation cycles, are available for download. The experiment data include a 6-hourly analysis for the three experiments, Control, ZTD, and ZTDGRA, for June and July 2021. All the files are stored in Zenodo, a general-purpose open repository developed under the European Open Access Infrastructure for Research in Europe (OpenAIRE) program and operated by the European Organization for Nuclear Research (CERN). The access link is: https://doi.org/10.5281/zenodo.10276429.

*Author contributions.* The original draft of the study was written by RT, who also conducted the formal analysis and experiments. FZ and RT collaborated to modify the WRFDA code and develop the gradient operator. JW and GD supervised the project, acquired funding, and reviewed and edited the paper.

*Competing interests.* The contact author has declared that none of the authors has any competing interests.

*Acknowledgements.* The research project is funded by the German Research Foundation (DFG) and titled "Exploitation of GNSS tropospheric gradients for severe weather Monitoring And Prediction (EGMAP)."The ECMWF conventional datasets for the DA study in this research were provided by Dr. Thomas Schwitalla from our collaborative institution, the Institute of Physics and Meteorology, University of Hohenheim, Stuttgart. Zenodo (https://zenodo.org/) is used to store the model code and simulation data.



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



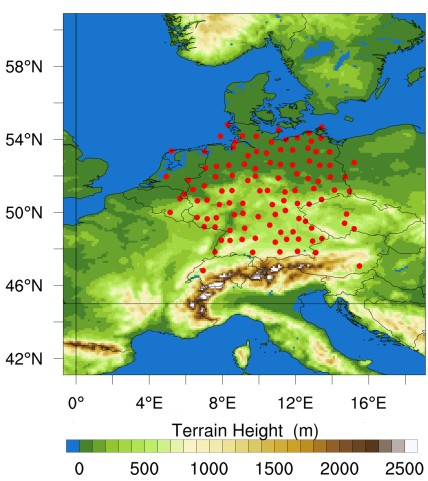

**Figure 1.** The WRF model domain at a horizontal resolution of $0.1°$ ($\approx 11$ km) with orography and the locations of the GNSS stations.





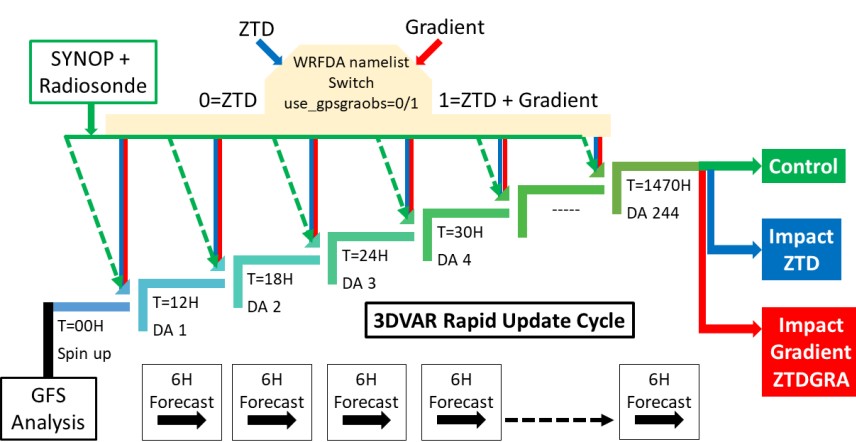

**Figure 2.** Schematic of the 3DVAR rapid update cycle initialized from the GFS analysis. A spin-up of 12 hours was performed until 00 UTC on June 01, 2021. Three experiments with different setups are performed: Control run (green) assimilating conventional data, ZTD run (blue) assimilating ZTDs on top of the control run, and ZTDGRA run (red) assimilating ZTD and gradients on top of the control run. The WRFDA namelist switch use_gpsgraobs has 0 for ZTD assimilation and 1 for ZTD and gradient assimilation.





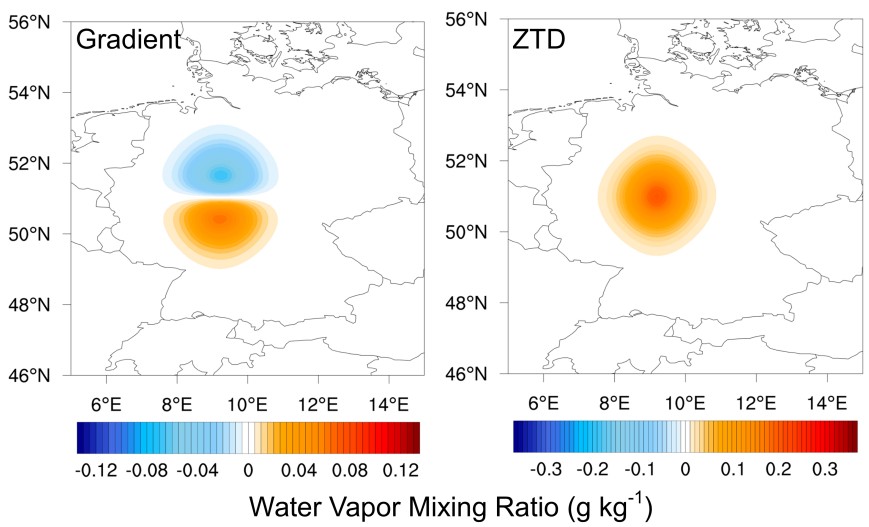

**Figure 3.** Single observation test spatial plot of gradient and ZTD. A N-gradient increment of -1.0 mm and an error of 1 mm is applied for the gradient SOT on the left. A ZTD increment of 1 cm and an error of 1 cm is applied on the right.



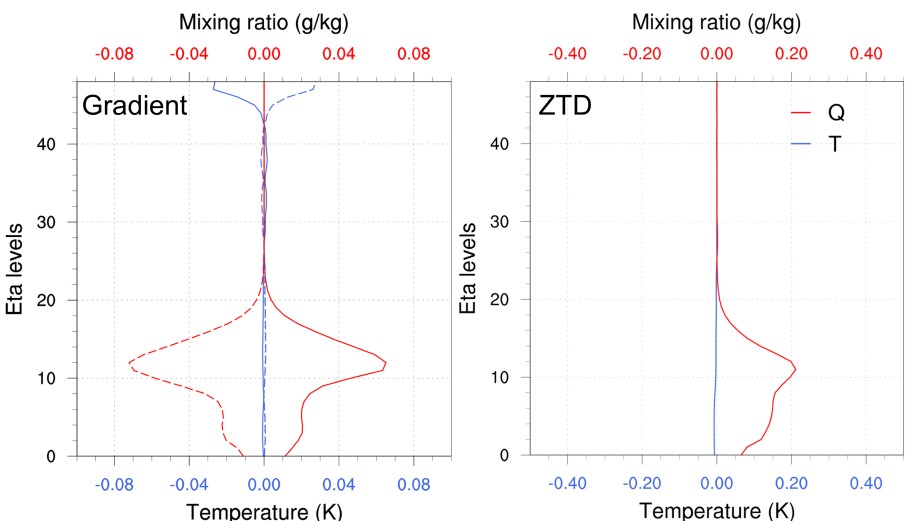

**Figure 4.** The vertical profile of the SOTs with the same settings as Figure 3. The gradient SOT on the left shows the positive and negative (dash) lobes of the impact.



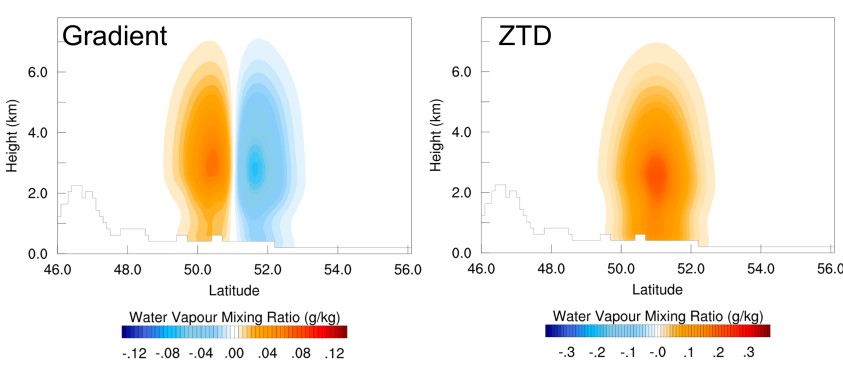

**Figure 5.** The vertical cross section of the SOTS with the same settings as Figure 3.



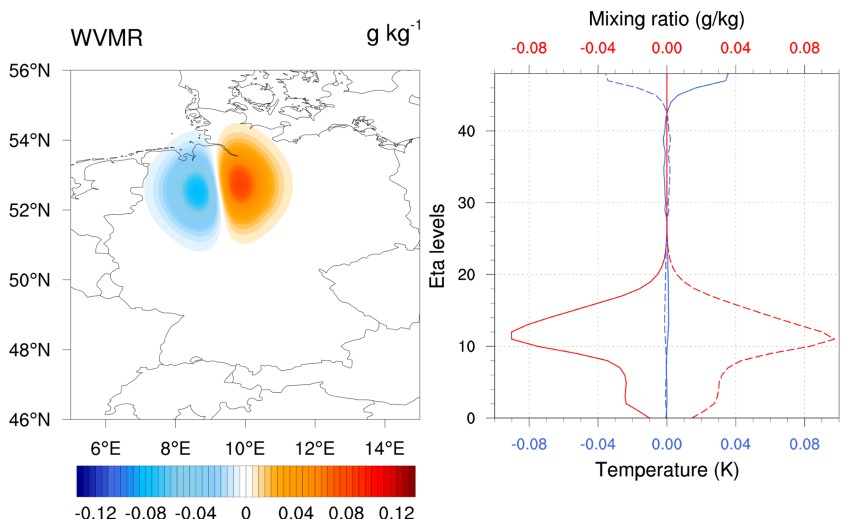

**Figure 6.** A real observation SOT with gradient observations: The east gradient component equals 0.497 mm, and the north gradient component equals0.099 mm. The observation error value for the SOT is chosen to be 0.65 mm which is the standard error used for this study.



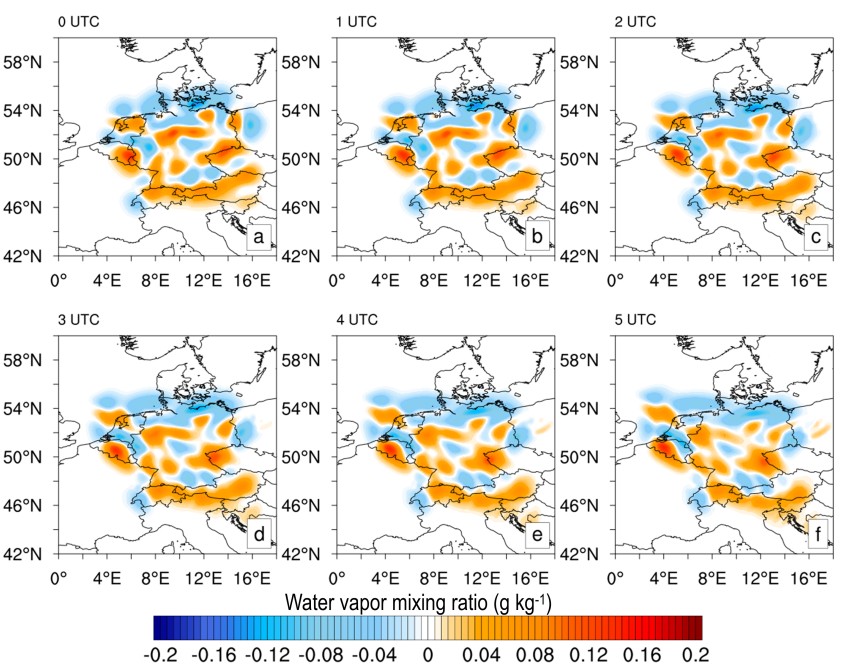

**Figure 7.** Spatial analysis and forecast differences: ZTDGRA minus ZTD experiment. The figure shows the exclusive impact of the gradient observation for the first DA cycle at 00 UTC (analysis) and the five-hour forecast from the analysis.



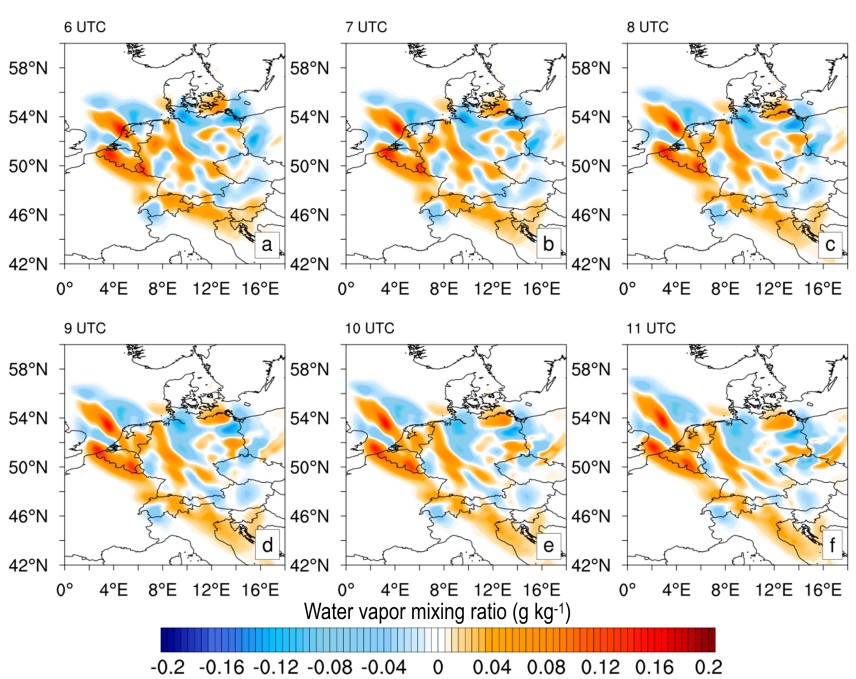

**Figure 8.** Same as Figure 7 but for the second DA cycle.





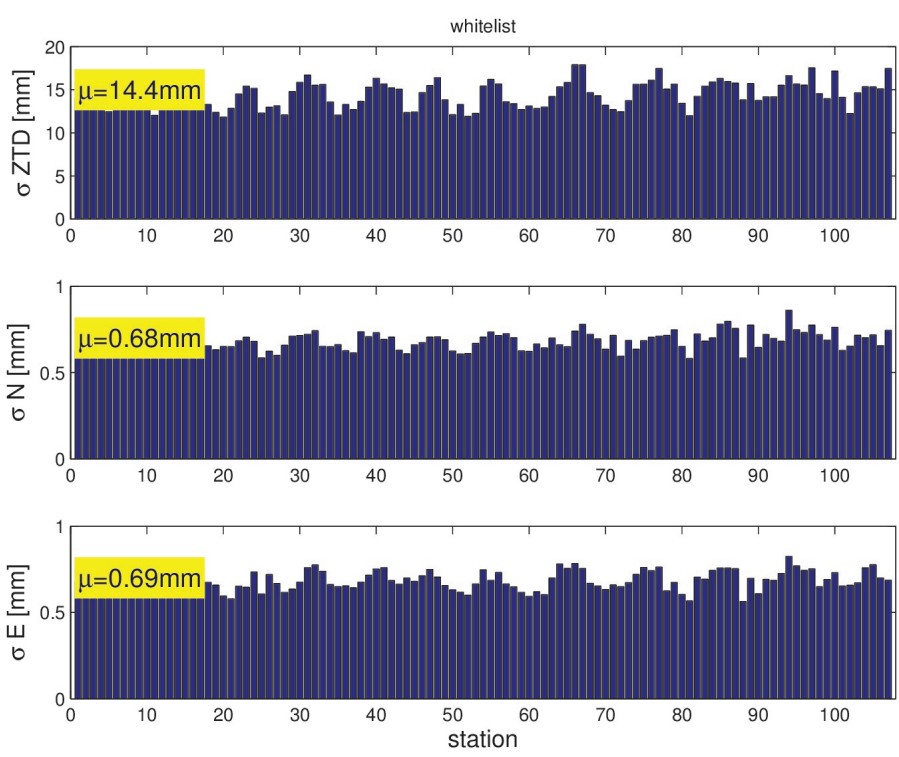

**Figure 9.** Control run: The station specific RMSE of the ZTD, North and East components (whitelisted stations). The average RMSE is given by $\mu$.





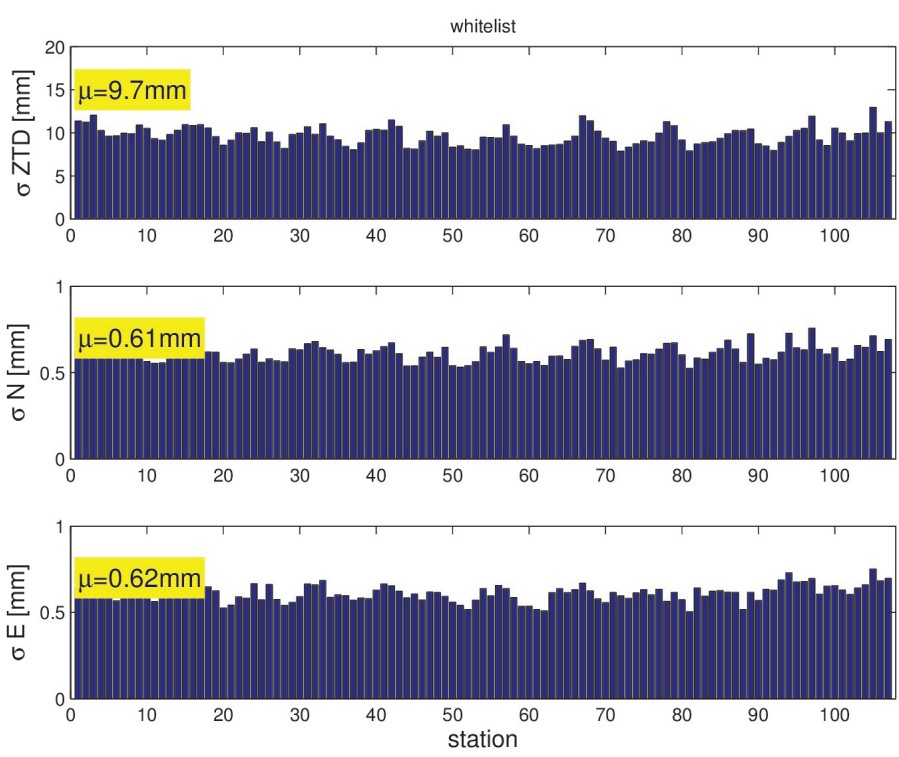

**Figure 10.** ZTD run RMSE of the ZTD and gradients: North and East components with respect to the whitelisted stations. The mean RMSE is given by $\mu$.



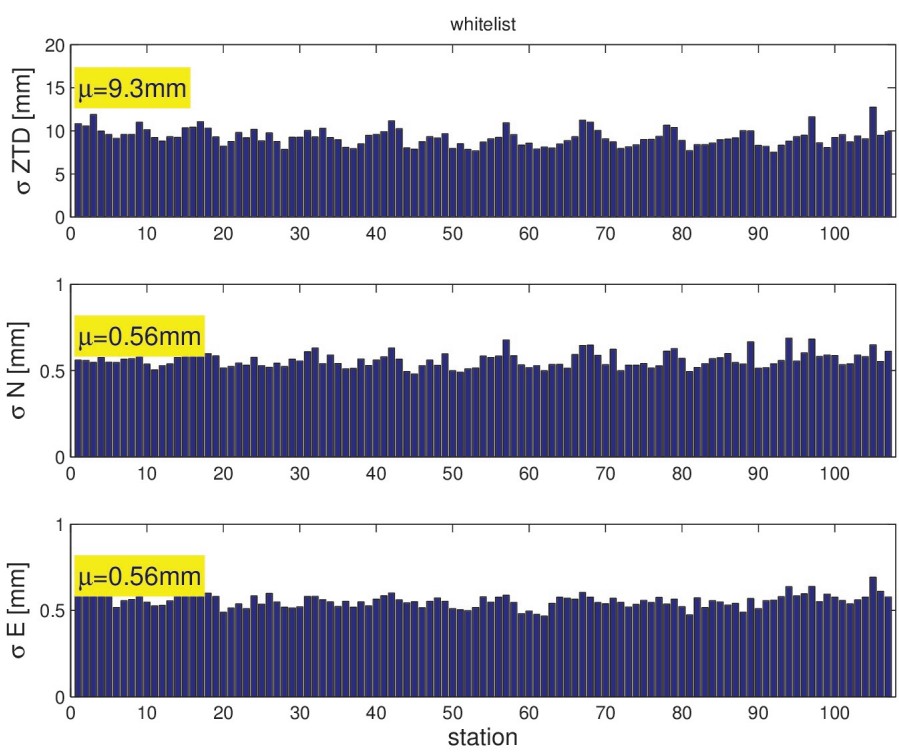

**Figure 11.** ZTDGRA run RMSE of the ZTD and gradients: North and East components with respect to the whitelisted stations. The mean RMSE is given by $\mu$.



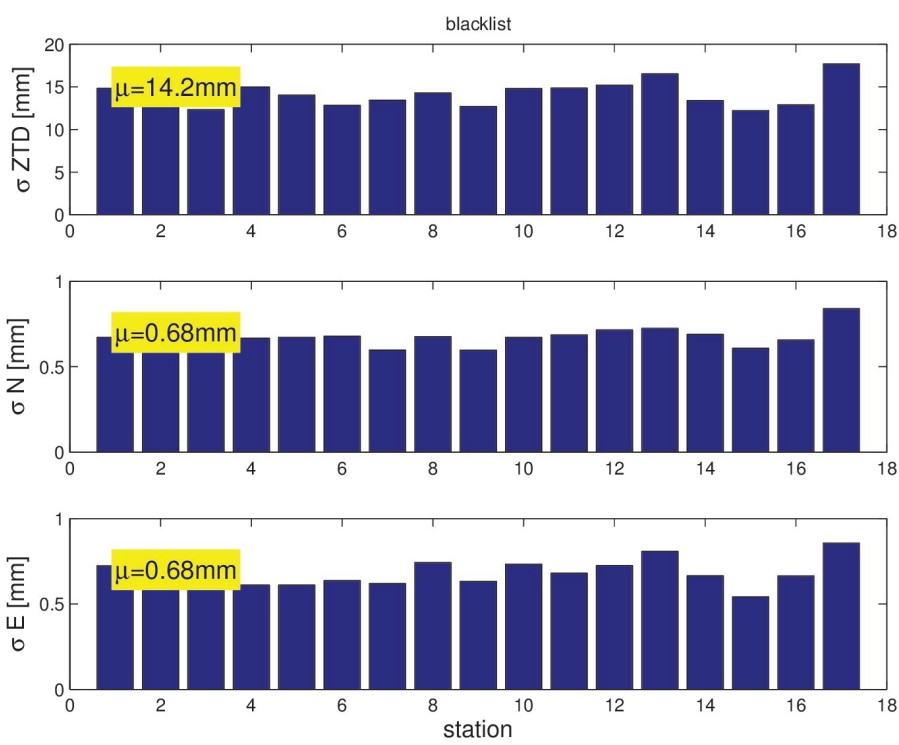

**Figure 12.** Control run RMSE of the ZTD and gradients: North and East components with respect to the blacklisted stations. The mean RMSE is given by $\mu$.





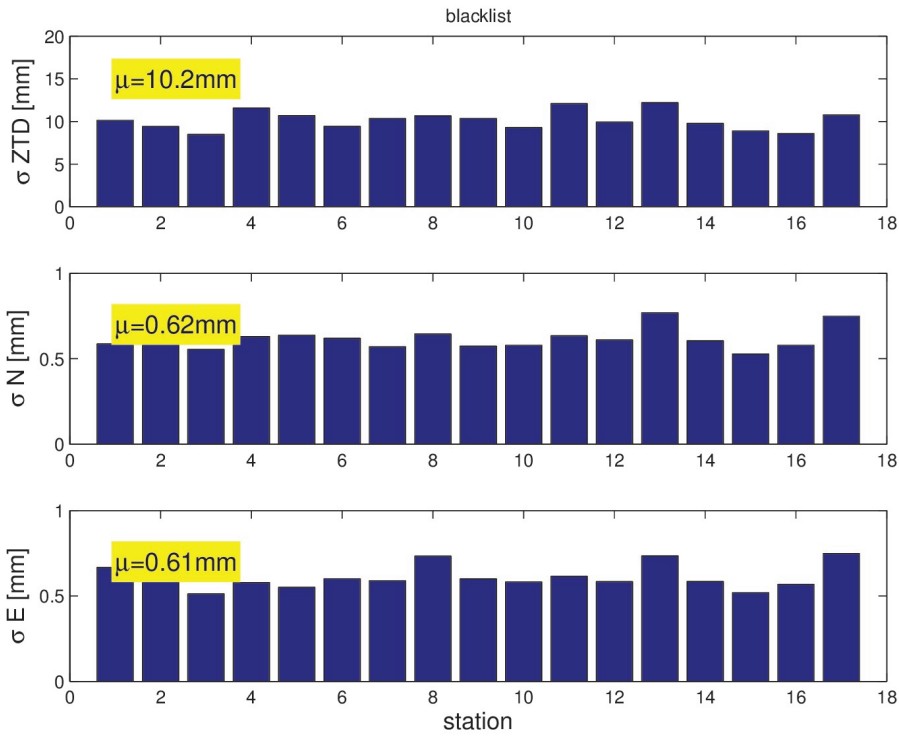

**Figure 13.** ZTD run RMSE of the ZTD and gradients: North and East components with respect to the blacklisted stations. The mean RMSE is given by $\mu$.



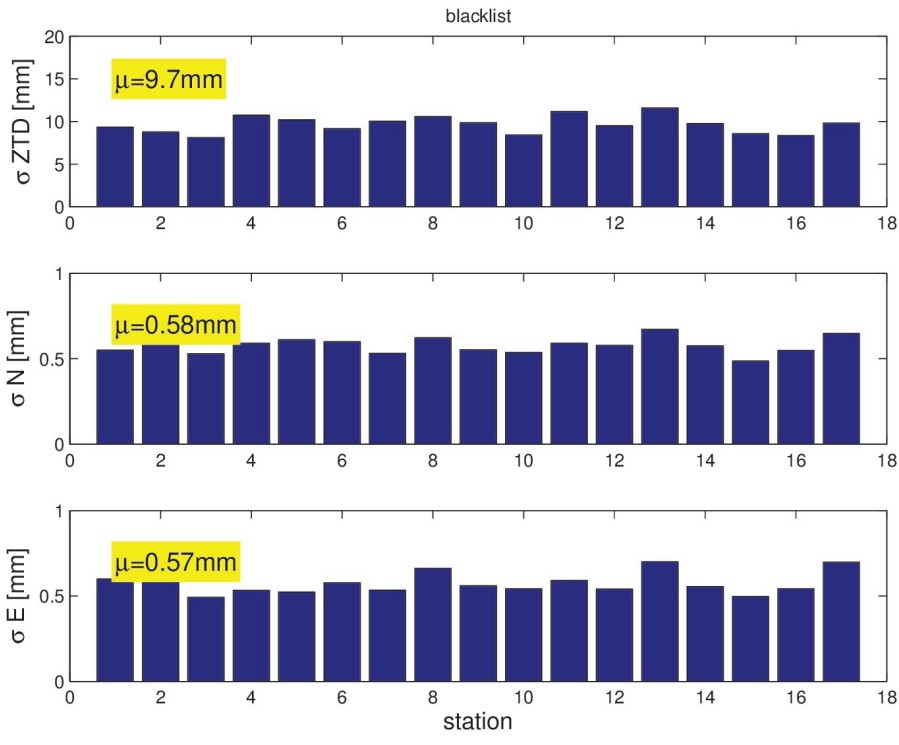

**Figure 14.** ZTDGRA run RMSE of the ZTD and gradients: North and East components with respect to the blacklisted stations. The mean RMSE is given by $\mu$.





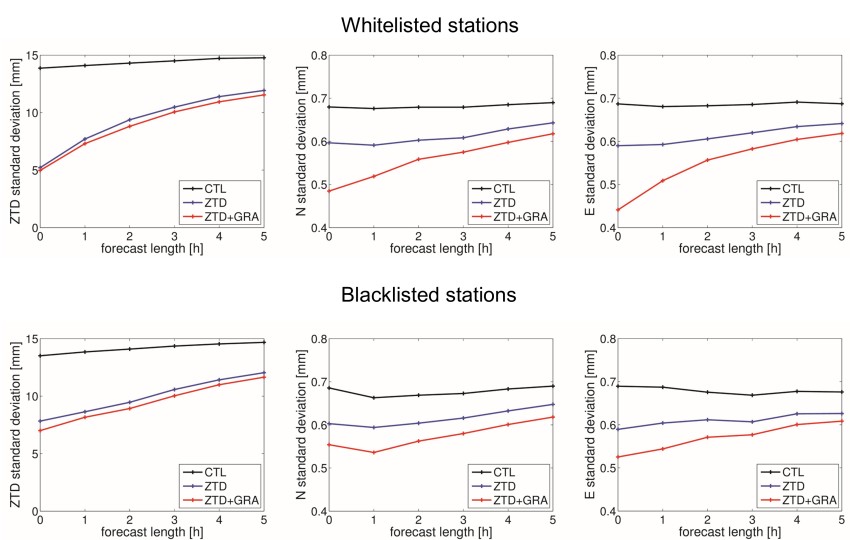

**Figure 15.** Average impact with respect to forecast lead times from analyses for two months of simulation. The Control run (black), ZTD run (blue), and ZTDGRA run (red) is shown for the whitelisted stations (top row) and for the blacklisted station (bottom row).





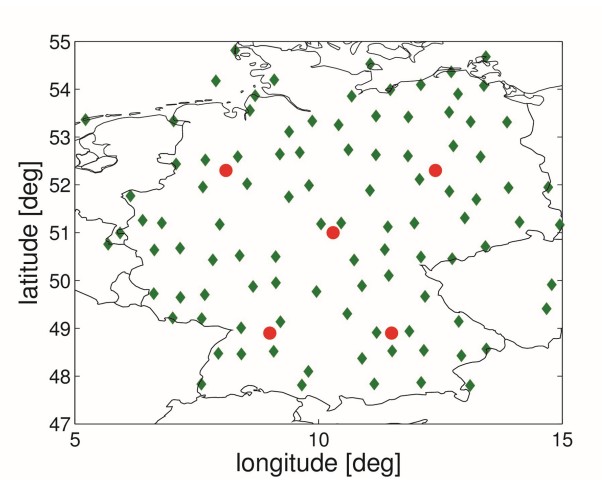

**Figure 16.** ERA5 comparison domain. The station map and the selected points for comparison of the ERA5 data with the analyses.



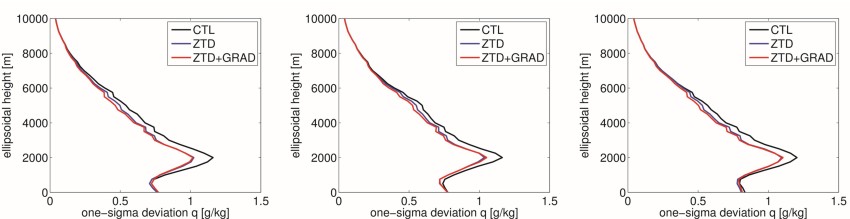

**Figure 17.** ERA5 profile comparison. The statistics of 1220 profiles for the Control run (black), ZTD run (blue), and ZTDGRA run (red) is shown for the analyses (00 UTC) on the left-most, 3-hour time-lead on the middle, and 5-hour time-lead on the right-most.



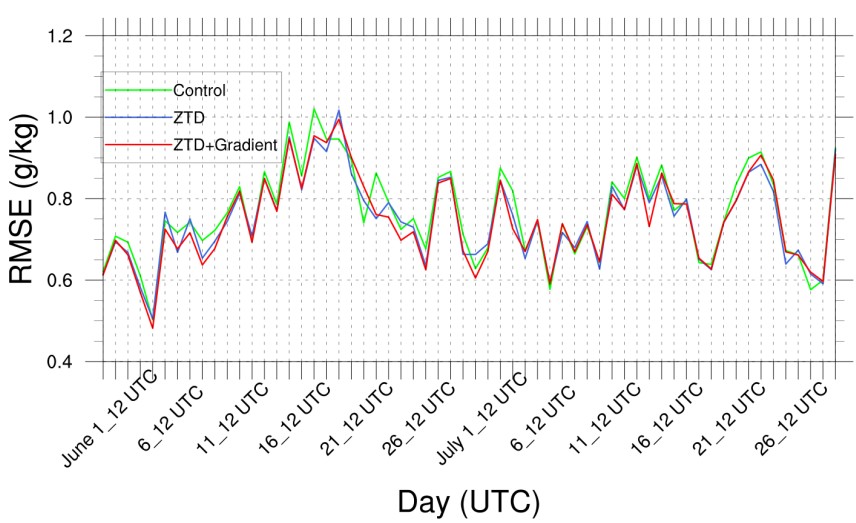

**Figure 18.** Comparison of the 12-hour forecasts from the analyses to the radiosonde profiles (around 38 radiosondes per epoch).



**Table 1.** Model Physics

| Physics | WRF options |
|---|---|
| Long wave radiation | RRTMG (Iacono et al., 2008) |
| Short wave radiation | RRTMG (Iacono et al., 2008) |
| Cloud microphysics | Thompson scheme (Thompson et al., 2008) |
| Cumulus scheme | Grell and Freitas scheme (Grell and Freitas, 2014) |
| Planetary boundary layer | YSU scheme (Hong et al., 2010; Hong and Lim, 2006) |
| Land surface scheme | Unified Noah land-surface model (Chen and Dudhia, 2001) |