# Peer review of "Assimilation of GNSS Tropospheric Gradients into the Weather Research and Forecasting Model Version 4.4.1"

_Geoscientific Model Development, 2023_

## Author Comment (AC1)

I congratulate the authors on a nicely written, interesting article on the assimilation of GNSS ZTDs and ZTD gradients into WRF using a fast observation operator, avoiding time consuming ray tracing. It has high value for the GNSS meteorological community. Being a first test one can always ask for more details and work, but I advocate publication with just minor adjustments in order to make it quickly available for others to use. It will be interesting to see in the future whether higher resolution NWP shows improved impact, and whether the benefit is still clear in NWP using more types of standard meteorological observations.

We thank the reviewer for taking the time to review the manuscript and for the kind words. Thank you very much for understanding the scientific motive behind our research. We want to share the GNSS gradient operator with the GNSS meteorological community so that we can optimize the use of gradients and figure out ways to use this abundant observation type in the operational forecast centers worldwide. We hope the readers can test the operator and make any improvements to the existing version.

Detailed comments:

Don't use dots as a sign for multiplication, like in equation 3. When you have defined \phi as a function, it is clear that \phi \delta z means multiplication.

Thanks for the comment. This has now been corrected.

You use a limited set of standard meteorological observations in your DA. SYNOP will be available at all 4 assimilation times, but radiosondings are more rare at 06 and 18 UTC, than 00 and 12 UTC. Please specifiy how large the difference is in your area.

We agree with the reviewer that radiosondes are rare at 06 and 18 UTC compared to 00 and 12 UTC. To specify the difference in our domain, we have included a table below that indicates the average number and type of observations assimilated based on the respective timesteps for the two months. We have now incorporated the details in the revised manuscript. We would also like to point out that we had Tropospheric Airborne Meteorological Data Reporting (TAMDAR) observations along with the surface stations and radiosondes. We did not mention this in the manuscript by mistake. We take this opportunity to correct it.

| Assimilation Timestep | SYNOP | Radiosonde | TAMDAR | GNSS | Total |
|---|---|---|---|---|---|
| 00 UTC | 1285 | 31 | 80 | 100 | 1496 |
| 06 UTC | 1327 | 20 | 2196 | 102 | 3645 |
| 12 UTC | 1328 | 60 | 2275 | 103 | 3766 |
| 18 UTC | 1318 | 12 | 1752 | 103 | 3185 |

In figure 2, what does the shorter and shorter vertical green, blue and red lines indicate? Specify explicitly whether the 3 simulations run independently, ie. the ZTD and ZTD+gradients are assimilated in first guesses based on ZTD only and ZTD+gradients, not into the control first guess.

Thank you for pointing out the ambiguity in the figure. We agree that the colored arrows in the figure can be misleading and may confuse the readers. The ZTD and ZTD + gradients are two different experiments independent from the Control experiment. They do not use control first guess. This is now clarified and explained in the revised manuscript. Also with reference to comments from the second reviewer we have added an additional experiment with only gradient assimilation (GRA). The new figure is attached below.

[Figure]

**Figure 1.** Schematic of the 3DVAR rapid update cycle initialized from the GFS analysis. A spin-up of 12 hours was performed until 00 UTC on June 01, 2021. Four experiments with different setups are performed: Control run (black) assimilating conventional data, ZTD run (purple) assimilating ZTDs on top of the control run, ZTDGRA run (red) assimilating ZTD and gradients on top of the control run, and GRA run (green) assimilating gradients on top of the control run. The WRFDA namelist switch use_gpsgraobs has 0 for ZTD assimilation and 1 for ZTD and gradient assimilation.

In figure 4, what is the location of the "lobes" where you obtain the profiles (is it for example a certain distance from the location of the GNSS sites?

We thank the author for the comment. We have now explained how we obtain the profiles in the revised manuscript.

The location of the lobes are calculated through the following steps:

1. Average the analysis over the vertical levels between 2 km and 4 km, where the gradient observations have the maximum influence.

2. Determine the latitude and longitude of the point in the domain where the absolute value of the water vapor mixing ratio is maximum.
3. Through step 2, we get two maximum value points on both sides of the gradient observation location. The profile is derived from these two locations; hence, we get the lobes.

What is shown in figure 18?  Is it the average rmse of the mixing ratio up through the radiosonde profile? That might be dominated by mixing ratios at certain heights. Or is it rmse at a certain level?

Thank you for the comment. We have now incorporated the explanation of this comment in detail in the manuscript for clarity.

Figure 18 shows the average RMSE of the mixing ratio up through the radiosonde profile. Since radiosonde profiles are usually available at irregular heights, we interpolated the model profiles to that of the radiosondes. Hence, the model profiles of the 12-hour forecast from the analyses were compared with the radiosonde profiles at that time instant to calculate the RMSE. We compared the forecasts valid at 12 UTC (forecasted from analyses at 00 UTC) with the radiosondes at 12 UTC, since there were more radiosondes during that time, to get the statistics. There were a minimum of 38 radiosondes available for comparison. The new updated figure (Figure 2) with the radiosonde RMSE forecast comparison is shown below. The new run GRA is also included in the updated figure.

[Figure]

**Figure 2.** Water vapor mixing ratio comparison of the 12-hour forecasts from the analyses to the radiosonde profiles (around 38 radiosondes per epoch). The mean RMSE (g/kg) of the respective runs for the two months are specified in the legend.

---

## Author Comment (AC2)

General Comment:

The authors have implemented the GNSS tropospheric gradient operator into WRFDA version 4.4.1 and conducted single observation tests with ZTD data and tropospheric gradients. Three experiments, employing a rapid-update cycle throughout June and July 2021, were carried out to investigate the impact of assimilating tropospheric gradients. The analyses and simulations have been verified against GNSS data from 100 stations, ERA5 reanalysis, and radiosondes. It is nice that the authors integrated the new operator into the WRFDA data assimilation system, and the manuscript is well-written.

We thank the reviewer for the valuable time to evaluate the manuscript and appreciate the kind words. Our motive is to encourage the use of the code by the GNSS Meteorology community so that we can incorporate improvements in the future.

However, it is noted that the comparisons are primarily based on the control run with limited observations involved. Therefore, the impacts of additional observations overlaid on the control run could be overestimated. Specific comments are provided as follows.

Specific comments:

This study implemented the tropospheric gradient operator atop the GNSS ZTD modules in the WRFDA system. While the authors stated that the manuscript aims to test the functionality of the operator and assess the relative impact of tropospheric gradients, it is noted that the control run assimilated with surface stations and radiosondes only is limited and insufficient for the impact study. The comparing experiments (ZTD and ZTDGRA) added ZTD and tropospheric gradient data on top of the two types of observations in the control run, which could potentially enlarge the data impacts of ZTD and tropospheric gradient. A suggestion is to incorporate most of the observations adopted in the operational model for the control run. This aligns with the goal of the EGMAP, as mentioned in lines 97-98.

Thank you very much for the elaborative comment. First of all, we want to make a correction in the article text that the types of observations used for the assimilation experiment had Tropospheric Airborne Meteorological Data Reporting (TAMDAR) observations, too, along with the surface stations and radiosondes. We failed to mention this in the text since we had two versions of the impact study, of which this one is the refined version, which included TAMDAR too to incorporate more upper air observations apart from radiosondes. We have depicted a table showing the average number of observations assimilated at each time step in the responses to the first reviewer.

We agree with the reviewer that our motive was to test the functionality of the operator and assess the relative impact of the tropospheric gradients. However, we do not agree with the statement that the control run assimilated with surface stations and radiosondes is insufficient for the current impact study. Our research introduces the capability to incorporate a new observation type, i.e., tropospheric gradients, which have yet to be utilized by the operational

forecast community or research groups. We are trying to show how the gradients impact the analyses, and that is why we wanted to keep only the critical observation types, that is, the surface stations and the radiosondes, in the control run. We acknowledge the reviewer's reasoning completely that significant improvement may not be visible if we incorporate observations like satellite radiances in the control run.

Nevertheless, we still expect a slight improvement if we add ZTDs and gradients on top of all the conventional observations. As a pioneering research using GNSS gradients, the first step through this article was to assimilate gradient observations through an observation operator. Quantifying the improvement made by gradients for the prediction of severe weather for operational purposes, as mentioned in EGMAP, is beyond the scope of this manuscript and will be a topic for another article with the use of an ensemble data assimilation system.

In addition, incorporating more observations into the data assimilation usually benefits the model's initial analysis. The study conducted a long period of cycling data assimilation within a model domain that covered a larger region than the assimilated observations' coverage. Is there a specific reason for not utilizing all the observations within the model domain?

We appreciate the reviewer for raising the question. We will further clarify the point in the manuscript.

We had approximately 380 globally distributed GNSS station provided by the GFZ. We created a homogenized array of observations within the region of interest (Germany), i.e., we removed collocated stations, clusters of stations etc. As stated in the manuscript we only selected GNSS stations with data availability above 75% Hence, we finally had slightly more than 100 GNSS stations within Germany for the assimilation experiment.

To evaluate the impact of tropospheric gradient data, the study compares the difference between ZTD and ZTDGRA rather than comparing the assimilation without ZTD data (i.e., only the conventional observations and tropospheric gradient) with the control run. On the other hand, section 4.3.1 discussed that the ZTD run adjusted not only the ZTDs but also the tropospheric gradients. When assimilating both ZTD and tropospheric gradient observations, would it be overweighting the effects of the tropospheric gradient? Could you further elaborate on the interaction and influence as both data are assimilated simultaneously?

We thank the reviewer for the valuable comment. We have now elaborated the article with one more run added to the list of experiments. We have now incorporated a Gradient-only run to show the impact of gradients exclusively in the revised manuscript. Through this run, we want to point out that ZTDs are not providing a weightage to improve the gradients. ZTD plays a role in the improvement of the gradient and vice versa. We have enhanced the manuscript with new plots for comparing four experiments. Please refer to the response to the following comment for the illustration with the new experiment run "GRA" included, which is the gradient-only assimilation.

Figures 9-11 indicate similar information. Merge the three figures into one would be clearer for comparison. For example, display the RMSE of ZTD for the control run, ZTD run, and ZTDGRA run by three curves on one panel. Similar processes for RMSEs of the North and East components on the second and third panels. The same suggestion is for Figures 12-14.

Thank you for the comment. In the updated manuscript, we have now combined all the similar plots into one plot. Figure 1, shown below, replaces figures 9, 10, and 11 in the manuscript, depicting the whitelisted station-specific RMSE of the ZTD and gradient North and East components for different runs, followed by a table summarizing the mean. Figure 2 (replacement for figures 12, 13 and 14) and Table 2 shows the comparison for the blacklisted stations.

[Figure]

**Figure 1.** The station specific RMSE of the ZTD, North and East components (whitelisted stations): Control (black), GRA (green), ZTD (purple), and ZTDGRA (red).

**Table 1.** Comparison of the mean (μ) of station specific RMSE of ZTD and Gradients (whitelisted stations).

| Mean (μ) | ZTD | North Gradient | East Gradient |
|---|---|---|---|
| Control | 14.4 | 0.68 | 0.69 |
| GRA | 12.4 | 0.58 | 0.57 |
| ZTD | 9.7 | 0.61 | 0.62 |
| ZTDGRA | 9.3 | 0.56 | 0.56 |

[Figure]

**Figure 2.** The station specific RMSE of the ZTD, North and East components (blacklisted stations): Control (black), GRA (green), ZTD (purple), and ZTDGRA (red).

**Table 2.** Comparison of the mean (μ) of station specific RMSE of ZTD and Gradients (blacklisted stations).

| Mean (μ) | ZTD | North Gradient | East Gradient |
|---|---|---|---|
| Control | 14.2 | 0.68 | 0.68 |
| GRA | 12.4 | 0.58 | 0.57 |
| ZTD | 10.2 | 0.62 | 0.61 |
| ZTDGRA | 9.7 | 0.58 | 0.57 |

[Figure]

**Figure 3.** ERA5 profile comparison. The statistics of 1220 profiles for the Control run (black), GRA (green), ZTD run (purple), and ZTDGRA run (red) is shown for the analyses (00 UTC) on the left-most, 3-hour time-lead on the middle, and 5-hour time-lead on the right-most.

[Figure]

**Figure 4.** Average impact with respect to forecast lead times from analyses for two months of simulation. The Control run (black), GRA run (green), ZTD run (purple), and ZTDGRA run (red) is shown for the whitelisted stations (top row) and for the blacklisted station (bottom row).

Figures 3 and 4 clearly show that ZTDs do not provide a weightage to improve the gradients. The gradient observations alone have an individual impact on the analyses. Both the observations, ZTDs, and gradients improve each of the variables in the analyses. We hope the explanation with the GRA run provides clarity to the previous comment.

In WRFDA, it has converted geometric height to the geopotential height for GNSS refractivity, not as the description in the manuscript to ignore the conversion. It can be found in da_fill_obs_structures.inc. Ignoring the conversion could result in some errors, particularly at higher altitudes (Scherlllin-Pirscher et al. 2017).

We work with the geometric height. I.e. we convert the geopotential height (stored in 'grid%xb%h' or for the mpi-routine 'glob_h') to the geometric height. However, when we approximate refractivity above the model top utilizing the hydrostatic equation we use for simplicity the geometric height and not the geopotential height (see eq 10 in the manuscript). This approximation is not problematic as the horizontal refractivity gradients at height altitudes are small and hence the contribution to the tropospheric gradient (integral) is negligible. For details on the implementation see 'da_get_innov_vector_gpsztd.inc'.

Scherllin-Pirscher, B., A. K. Steiner, G. Kirchengast, M. Schwärz, and S. S. Leroy (2017), The power of vertical geolocation of atmospheric profiles from GNSS radio occultation, J. Geophys. Res. Atmos., 122, 1595–1616, doi:10.1002/2016JD025902.

Typos:

Line 241. The NMC method is widely used for generating B by … --> Please revise B to bold B.

Thanks. This is now corrected.

Figure 6. A real …. component equals0.099 mm. --> Please add a blank between equals and 0.099 mm.

Thanks. This is now corrected.

---

## Author Response (AR2)

**Manuscript ID**           :        **gmd-2023-202**
**Manuscript title**        :        **Assimilation of GNSS Tropospheric Gradients into the**
                                     **Weather Research and Forecasting Model Version 4.4.1**

Corresponding author        :        Dr Rohith Muraleedharan Thundathil
Affiliation                 :        Technische Universität Berlin;
                                     GFZ German Research Centre for Geosciences Potsdam
Email                       :        r.thundathil@tu-berlin.de

Dear Editor,

Once again, we thank you for organizing our manuscript discussion in the GMD. We really appreciate the Editor's choice of the appropriate reviewers and speeding up the review process. We also thank the reviewers for their valuable comments, which could help us improve the manuscript.

The point-by-point response to the Editor's comments will follow from the next page.

Best regards,

Rohith Thundathil

Dear authors,

Thank you for qualitatively addressing comments raised by reviewers. I support the manuscript for publication after a few minor technical corrections are applied.

We thank the Editor for the prompt and supportive response concerning our manuscript. We are delighted to hear that our past revisions were sufficient for publication in GMD.

Please take a look at GMD guidelines to make necessary technical corrections:

https://www.geoscientific-model-development.net/submission.html

such as:

- Please make several changes regarding abbreviations:

(a) Please use a consistent abbreviation for "WRF Data Assimilation" ("WRFDA") throughout the manuscript (currently you employ both "WRFDA" and "WRF DA" - the latter appears for example in Lines 14, 478).

Thanks for pointing it out. We have now corrected it throughout the manuscript.

Please see the Lines: 14, and 487.

(b) Abbreviation "TAMDAR" is used (e.g., Lines 253, 256) before it is defined in Line 268; - please correct this.

Thanks. This has now been corrected.

Please see the Line: 255.

(c) Given the fact that you define "European Centre for Medium-Range Weather Forecasts (ECMWF)", I would also advise you to define a few other abbreviations such as "HARMONIE" (which might be less known than ECMWF).

Thank you for the comment. We have now expanded a few abbreviations in the manuscript.

Please see the Lines:

45: "Action de Recherche Petite Échelle Grande Échelle (ARPEGE)"

49: "High Resolution Limited Area Model (HIRLAM) – Aire Limitée Adaptation Dynamique Développement International (ALADIN) Research on Mesoscale Operational NWP in Euromed (HARMONIE) - Applications of Research to Operations at Mesoscale (HARMONIE-AROME)"

80: "European Reference Frame (EUREF)"

(d) "TEMP" should be defined.

Thank you for the comment. TEMP has now been defined.

Please see the Line 268: "TEMP is a collection of alphanumerical codes established by the WMO. These codes represent upper air soundings obtained through weather balloons launched from either land or sea level. These observations report on the weather conditions in the upper regions of the atmosphere. In WRFDA, FM-35 TEMP is the observation code used to identify radiosonde observations launched from land, and FM-36 for those launched from ships."

(e) Given a large number of abbreviations appearing throughout the manuscript (which can be difficult to read), I would advise you to omit the introduction of abbreviations which you don't use anywhere later in the manuscript such as: "World Meteorological Organization (WMO)", "Global Telecommunication System (GTS)", "Radiosonde (RS) measurements", ... These could be simply "World Meteorological Organization", Global Telecommunication System, "Radiosonde measurements", ...

Thank you for the comment. The abbreviations which are not used anywhere else in the manuscript are omitted.

- Units must be written exponentially (e.g. W m$^{-2}$). Thus please correct the way you write units at several places within the manuscript (e.g., Figures 4, 5, 6, 13, 14).

Thanks. This has now been corrected.

- Labels of figure panels must be included with brackets around letters being lower case (e.g. (a), (b), etc.).

Thanks. This has now been corrected.

- Dates should be written in the following form: 25 July 2007 (dd month yyyy).

Thanks. This has now been corrected.

Please see Lines: 261, 357, 471, and also Fig 14.

- Line 48: "NWP forecasting" -> "NWP" to avoid redundant words.

Thanks. This has now been corrected.

Please see Line: 48.

Thank you in advance for your cooperation.

Best regards,

Nina Črnivec, Topic Editor, Geoscientific Model Development